# Wolfram Syndrome Type 2: A Systematic Review of a Not Easily Identifiable Clinical Spectrum

**DOI:** 10.3390/ijerph19020835

**Published:** 2022-01-12

**Authors:** Francesco Maria Rosanio, Francesca Di Candia, Luisa Occhiati, Ludovica Fedi, Francesco Paolo Malvone, Davide Fortunato Foschini, Adriana Franzese, Enza Mozzillo

**Affiliations:** Regional Center of Pediatric Diabetes, Department of Translational Medical Sciences, Section of Pediatrics, Federico II University, Via Sergio Pansini 5, 80131 Naples, Italy; francescomaria.rosanio@unina.it (F.M.R.); francesca.dicandia@unina.it (F.D.C.); luisa.occhiati@unina.it (L.O.); ludovica.fedi@unina.it (L.F.); francescopaolo.malvone@unina.it (F.P.M.); davidefortunato.foschini@unina.it (D.F.F.); franzese@unina.it (A.F.)

**Keywords:** Wolfram syndrome type 2, CISD2, DIDMOAD, genetic diabetes, optic atrophy, hearing loss, bleeding disorders, gastrointestinal ulcers

## Abstract

Background: Wolfram syndrome (WS) is a rare autosomal recessive disorder that is characterized by the presence of diabetes mellitus, optic atrophy and hearing loss, all of which are crucial elements for the diagnosis. WS is variably associated with diabetes insipidus, neurological disorders, urinary tract anomalies, endocrine dysfunctions and many other systemic manifestations. Since Wolfram and Wagener first described WS in 1938, new phenotypic/genotypic variants of the syndrome have been observed and the clinical picture has been significantly enriched. To date, two main subtypes of WS that associated with two different mutations are known: WS type 1 (WS1), caused by the mutation of the wolframine gene (WS1; 606201), and WS type 2 (WS2), caused by the mutation of the CISD2 gene (WS2; 604928). Methods: A systematic review of the literature was describe the phenotypic characteristics of WS2 in order to highlight the key elements that differentiate it from the classic form. Conclusion: WS2 is the rarest and most recently identified subtype of WS; its clinical picture is partially overlapping with that of WS1, from which it traditionally differs by the absence of diabetes insipidus and the presence of greater bleeding tendency and peptic ulcers.

## 1. Introduction

Wolfram syndrome (WS) is a rare autosomal recessive neurodegenerative disorder that includes four cardinal features: diabetes mellitus (DM), optic atrophy (OA), diabetes insipidus (DI) and sensorineural hearing loss (SNHL) [1,2,3]. The old acronym of DIMOAD (diabetes insipidus, diabetes mellitus, optic atrophy and deafness) that was previously used to describe the syndrome appears now anachronistic: several other abnormalities (including neurological, psychiatric, urological, endocrinological, rheumatological disorders) have been described in association with the classical phenotype, making WS an extremely complex and varied condition.

WS was first described in 1938 by Wolfram and Wagener, who reported the classic association of diabetes mellitus and optic atrophy in four siblings of consanguineous parents [4]. In a subsequent eighteen-year follow-up, these four siblings developed deafness as well as neurological (atypical Friedreich’s ataxia) and urinary disorders (neurological bladder) [5].

Since then, numerous cases of WS have been reported and new clinical findings have been added to the spectrum of the syndrome, which has become extremely broad. Currently, two main subtypes of WS are known: Wolfram syndrome type 1 (WS1), caused by the mutation of the wolframine gene (WS1; 606201)—a transmembrane protein localized in the endoplasmic reticulum whose deficiency causes increased cellular stress and apoptosis, and Wolfram syndrome type 2 (WS2), caused by the mutation of the CDGSH iron-sulfur domain-containing protein 2 gene (CISD2 gene) (WS2; 604928) and first described in a Jordanian family [6].

WS is a very uncommon syndrome, with an estimated prevalence of approximately 1 in 770,000 [7]. This estimate is questionable; the prevalence would appear to be significantly higher in some countries, such as Lebanon where WS is thought to be responsible for up to 5.5% of the cases of juvenile-onset diabetes [8], and much lower in other countries as reported by a recent multi-center international study [9] in which WS is estimated to cause no more than 0.5% of juvenile-onset diabetes. This difference would seem to be partly related to the greater rate of consanguineous marriage in the Arab world. Unfortunately, due to the extreme rarity of the diagnosis, no epidemiological data are available about WS2.

The CISD2 gene is located on chromosome 4q24 and consists of 3 exons encoding the protein NAF-1 (nutrient deprivation autophagy factor-1), localized to the mitochondrial outer membrane and the endoplasmic reticulum [10]. CISD2 plays an essential role in ER and mitochondria membrane integrity. The ER is essential for cell survival and the correct folding of secretory proteins, surface receptors and integral membrane proteins. The sensitive folding environment of the ER can be perturbed by physiological processes. In these situations, if the ER exceeds its folding capacity, the balance of this delicate system can be disrupted with an accumulation of unfolded or misfolded protein inside the ER lumen. As a consequence, the cells undergo a condition that is known as ER stress, which activates a network of signaling pathways called the “unfolded protein response” (UPR) [11]. In pathological conditions, the UPR cannot reduce ER stress, unfolded/misfolded proteins accumulate in the ER lumen and the ER chaperones BIP (binding immunoglobulin protein) cannot bind their luminal domains, maintaining the unfolded or misfolded protein in an inactive state. The UPR performs its role by activating three signaling proteins: inositol-requiring protein 1 (IRE1), protein kinase RNA (PKR)-like ER kinase (PERK) and activating transcription factor 6 (ATF6). The activation of these transducers can culminate in cell death [12]. NAF1 belongs to the family of the NEET proteins, involved in iron management and the generation of reactive oxygen species (ROS). NAF1 deficiency seems to cause uncontrolled iron accumulation in the mitochondria and an excessive ROS generation with mitochondrial dysfunction, cellular damage and eventually apoptosis [13].

The main functions of the CISD2 gene can be summarized as follows: the regulation of calcium homeostasis between ER and cytosol; the regulation of autophagy by modulation of BECN1 activity; the regulation of oxidoreduction mechanisms and ROS production and the regulation of apoptosis by the modulation of calpain 2 activity (see Figure 1) [14].

WS1 is considered the traditional form of WS, in which the four cardinal elements (DM, OA, DI, SNHL) are (almost) always present. Compared to the classic form, WS2 patients show additional clinical features, such as greater bleeding tendency and peptic ulcer disease, which are uncommon in WS1. Conversely, diabetes insipidus appears to be rare/absent in WS2.

Recently, an autosomal recessive Wolfram-like syndrome was identified in a consanguineous Pakistani family: the three affected children each had sensorineural hearing impairment to all frequencies, diabetes mellitus and insipidus, gastrointestinal tract abnormalities and, unlike both WS1 and WS2, bicuspid aorta and no sign of optic atrophy. The latter is present in most published cases of WS1 and WS2, except in the case of Mozzillo et al. [10,15]. This syndrome was caused by a homozygous missense mutation on CDK13, a cyclin-dependent kinase-regulating gene transcription, alternative splicing of RNA and C-terminal domain [16].

WS is burdened with severe morbidity, early mortality and impaired quality of life. It is often difficult to obtain an early diagnosis because the clinical picture evolves over years, enriching itself with new elements. Often, these patients are treated as type 1 diabetes mellitus (this is usually the first clinical sign to emerge in the first years of life) and, only with the onset of vision or hearing impairment or other disorders, a syndromic form of diabetes is then suspected. An early diagnosis is essential in order to prevent complications and improve the quality of life; unfortunately, at the moment, no intervention has proven effective in slowing the progression of the disease.

This review aims to analyze the clinical phenotype of the genetically confirmed WS2 patients who have been described in the literature in order to highlight the key elements that differentiate this WS subtype from the classic form.

## 2. Materials and Methods

A systematic literature review was performed independently by these authors: L.O., F.P.M and D.F.F. The search terms were the following: “Wolfram syndrome” and “CISD2” or “wolframin” or “ERIS”. These systematic searches were conducted on electronic databases (PubMed, Scopus, Google Scholar, Embase, CENTRAL) and clinical trial registers (http://clinicaltrials.gov, accessed on 15 December 2021; www.controlled-trials.com, accessed on 15 December 2021).

Only papers on WS clinical findings within which the WS diagnoses were confirmed by genetic investigation were considered in this review. Animal studies were excluded. The articles that were selected for this literature review include those published from July 2001 to July 2021. The search included observational studies, prospective studies, cross-sectional studies, exploratory studies, case series, case reports and reviews. The papers were initially identified by the title and the abstract. Only documents showing that the WS diagnosis had been genetically confirmed were retrieved. Non-English manuscripts were not included in the research and review. Only full-text manuscripts were taken into account. The article selection process is shown in Figure 2.

## 3. Results

We collected data from 35 patients who each had a molecular diagnosis of WS2 (patient data are summarized in Table 1). No epidemiological data are available about WS2, probably due to the extreme rarity of the diagnosis.

The clinical spectrum of WS2 is only partially coincident with WS1, probably due to the different expressions of the genetic defect. WS may initially be mistaken for monogenic diabetes since non-autoimmune diabetes mellitus [17] is a primary characteristic. However, the simultaneous or subsequent association with other specific clinical features including vision, hearing, neurological, psychiatric, endocrinological, and urological disorders leads to suspicion of WS diagnosis.

In WS2 patients, diabetes insipidus and psychiatric disorders are not usually described. Still, the presence of some additional findings may suggest a diagnosis of defective platelet aggregation and peptic ulcers with a bleeding tendency.

### 3.1. Diabetes Mellitus

Diabetes mellitus is generally the first manifestation of WS (age at diagnosis: 6–10 years) followed by optic atrophy, which occurs later [18].

Diabetes mellitus is a nearly constant finding of WS: it is usually insulin-dependent, non-autoimmune, non-HLA-linked diabetes and, unlike type 1 diabetes mellitus, it is characterized by rare microvascular complications and a decreased tendency to develop ketoacidosis. Thanks to residual insulin secretion, diabetes in WS is characterized by a lower insulin requirement, lower levels of HbA1c and a milder clinical course than type 1 diabetes mellitus. For this reason, cases of WS2 using diabetes technologies (glucose sensors and insulin pumps) are not described by literature and self-blood glucose monitoring is performed by these rare pediatric patients who are affected by diabetes [19].

Unawareness of hypoglycemia can occur due to neurological dysfunctions [20,21].

Just as it does for WS1, diabetes represents one of the key elements for the diagnosis of WS2. Diabetes is present in all of the WS2 patients who have been described in the literature, with early onset in the first decade of life.

### 3.2. Optic Atrophy

Optic atrophy is usually diagnosed before 15 years of age and, along with non-autoimmune diabetes mellitus, is required for the diagnosis of WS. Optic atrophy is characterized by a progressive reduction of visual acuity and loss of color vision, leading to blindness. Unfortunately, no treatment is available to stop this visual degeneration [18]. Optic atrophy, without evidence of diabetic retinopathy, is the second key element that is indispensable for the diagnosis of WS. This was described in almost all of the collected cases; except in our case, in which the involvement of optic nerve was prevalently due to optic neuropathy rather than optic atrophy [10,15]. Generally, the diagnosis of optic atrophy or neuropathy follows the diagnosis of diabetes mellitus, but still occurs within the first two decades of life (age at diagnosis: 10–15 years) [18].

Less frequently comorbid ocular abnormalities include cataract, glaucoma, nystagmus, pigmentary and diabetic retinopathy and pigmentary maculopathy [22].

### 3.3. Sensorineural Hearing Loss

Slowly progressive sensorineural hearing loss occurs in about two thirds of WS patients and is usually diagnosed in the second decade of life (range 16–20 years). The clinical spectrum varies from a congenital form of deafness to a mild hearing loss. Regular audiological monitoring is suggested for the patients. Hearing aids and implants may be a solution for these patients [18]. In our case series, sensorineural hearing loss was found to be the third essential element of WS2 in terms of its prevalence; it was always present when investigated. However, deafness in WS2 does not always require a hearing aid as it is present in some cases but described only for high frequencies. Therefore, these are patients who carry out their daily activities normally but who have hearing problems only in noisy environments. Identifying this type of deafness is very difficult, hence diabetologists should consider a genetic form of diabetes [17] and request an audiogram.

### 3.4. Diabetes Insipidus

Diabetes insipidus is not a characteristic feature of WS2. Being more common in WS1, its association with WS2 was reported only in two cases [23,24]. Therefore, the absence of diabetes insipidus in WS2 cannot be considered pathognomonic; moreover, in these patients, partial forms of diabetes insipidus (which are difficult to diagnose) could be underestimated and, often, little investigated.

### 3.5. Endocrinological Disorders

The other typical endocrinological manifestations that are observed in WS include pituitary hypofunction, secondary hypogonadism (more common in males than females), delayed menarche, hypothyroidism and growth retardation [18].

A recent case report described asymptomatic hypoparathyroidism, osteomalacia and growth hormone (GH) deficiency in two siblings who are affected by WS2 whose diagnosis was not supported by a genetic investigation [25]. Except for these sporadic cases, no other endocrinological abnormalities are reported among the WS2 patients who are described in the literature.

### 3.6. Neurological Disorders

Neurological abnormalities are slowly progressive and occur later in about 60% of the WS patients (onset in the third–fourth decade of life). Magnetic resonance imaging demonstrates general brain atrophy, prominent in the cerebellum, medulla and pons [26] and brain stem as well as cranial nerve involvement [27]. The common symptoms are ataxia, central apnea, dementia, loss of the gag reflex, loss of olfaction, headache, myoclonus, epilepsy, nystagmus, gastroparesis, constipation and orthostatic hypotension [7].

Neurological abnormalities generally occur later in life and are therefore rarely described in patients of pediatric age. The most common manifestations include gastroparesis and neurogenic bladder with urinary incontinence. Only one case described cerebellar ataxia, myoclonic tremor, dysarthria, dysexecutive syndrome and a pseudobulbar affect in an adult patient with an abnormal brain MRI result [11]. In all of the case reports that have been analyzed, however, no mention is made of psychiatric disorders that do not seem to be typical of WS2.

### 3.7. Psychiatric Disorders

Psychiatric illness is frequently reported in WS: severe depression, psychosis, panic attacks, mood swings, suicidal behavior and impulsive verbal and physical aggression are significantly increased among WS1 patients. WS1 mutation carriers are also presumed to be predisposed to psychiatric illness [28]. No severe psychiatric disorders are reported among WS2 patients, except for mild to moderate mood disorders that are known to be related to chronic disease [24].

### 3.8. Urological Disorders

Neurological abnormalities may involve the urinary tract causing neurogenic bladder (hydroureter, urinary incontinence and recurrent infections are common signs of this diagnosis). Bladder dysfunction is very common in WS1 and WS2 (~90%), with progression to megacystis occurring over time [29]. A urodynamic examination is required in patients with these symptoms.

### 3.9. Gastrointestinal Disorders

It is not uncommon for gastrointestinal disorders to be the onset symptoms in WS2 [10,24]: bleeding gastrointestinal ulcers may be the first manifestation of WS2, which is diagnosed years later with the appearance of glucose metabolism impairment. In one case, the first clinical manifestation of WS2 was an acute abdomen requiring surgery due to intestinal malrotation associated with intussusception [10]. Gastrointestinal bleeding can also lead to investigation of the coagulation cascade, which is altered in almost all of the patients who are affected by WS2.

### 3.10. Bleeding Tendency

Al-Sheyyab investigated the bleeding tendency of 13 patients with WS2 versus a control group consisting of healthy and diabetic non-WS patients. Coagulation tests (investigating platelet count and aggregation with ADP, ristocetin and epinephrine; thrombin time; prothrombin time; activated partial thromboplastin time; clot retraction; factor VIII activity and von Willebrand factor antigen), performed for both the case and control groups, resulted normal. Eleven of the WS2 patients (85%) showed a prolonged template bleeding time and impaired aggregation with collagen when compared with both of the control groups [30]. Conversely, Mozzillo et al. described a platelet aggregation deficit in response to ADP (normal platelet aggregation in response to collagen, epinephrine and ristocetin) [10]. The pathogenesis of this hemorrhagic diathesis is unclear, but it would appear to be a peculiar feature of WS2 since no cases of abnormal bleeding have ever been reported in WS1.

### 3.11. Rheumatological Disorders

The association of rheumatological diseases with WS2 is infrequent; only Rondinelli et al. have described the presence of HLAB27 spondyloarthritis and Lofgren syndrome (a subtype of sarcoidosis) in an older and younger sister affected by WS2, respectively [24].

**Table 1 ijerph-19-00835-t001:** Major clinical features of Wolfram syndrome type 2 described in the literature.

Lead Author	CISD2 Mutation	Patients Characteristics	Onset Symptoms	DM	DI	OA	SNHL	ND	PD	UD	ED	IB-PU	PAD	Others
Al-Sheyyab, M., 2001 [30]	Sequencing data not available	13 (4 M, 9 F) Mean age: 14 years (6–37)										+	+ 11/13	
Danielpur, L., 2016 [31]	Homozygous mutation IVS1 6G C, p. E37Q	1 F 24 years at diagnosis		+	-	+	+	+ Neurogenic bladder	-	+	-	+		-
Mozzillo, E., 2014 [10]	Homozygous mutation Del(4)(q24)(chr4:103.803.602;103806759, 103.806818;103867350)x0	1 F 5 years at diagnosis	PU-IB	+	-	+ *	+	+ Gastroparesis Neurogenic bladder	-		-	+	+	Malrotation Intestinal intussusception
Pourreza, 2020 [32]	Homozygous mutation Missense variant c.310T > C (p.S104P) in exon 2	1 M 7 years at diagnosis	DM	+	-	+	-	+ Focal cortical gliosis Encephalomalacia				+		Meibomian gland dysfunction
Rondinelli, 2015 [24]	Homozygous mutation (cDNA: NM_001008388.4:c.103 + 1G > A, DNA Level: Chr4(GRCh37):g.103790345G > A)	2 F siblings <1 year at diagnosis	PU-IB	+	+ 1/2	+	+	+ Neurogenic bladder Abnormal nerve conduction study	+ Mood disorder	+ 1/2	+ Oligo-amenorrhoea	+	-	Lofgren syndrome Spondyloarthritis HLAB27+ Hypogammaglobulinemia Microcytic anemia
Rouzier, 2017 [11]	Homozygous mutation (c.215A > G; p.Asn72Ser)	1 M 45 years at diagnosis	DM	+ 8 years	-	+ 33 years	-	+ Severe neurological impairment Cerebellar ataxia Myoclonic tremor Dysarthria Pseudobulbar syndrome	-	+	-	-	-	Kidney cysts
Zhang, Y., 2019 [23]	Homozygous mutation (c.272_273del)	1 M 9 years at diagnosis	DM	+ 9 years	+ 11 years	+ 10 years	+ 11 years	Abnormal brain MRI Abnormal EEG	-	-	-	-	-	-
Riachi, M., 2019 [33]	Exon 3 deletion Chr4.del.83007475–113 025 264	1 M 19 years at diagnosis	DM	+	-	+	+	Developmental delay Cerebral atrophy		-		-	-	Ventricular defect
Ajlouni, K., 2002 [34]	Not available sequencing data	14 patients from 3 Jordanian families	DM/OA	+	-	+	+	Ataxia Abnormal nerve conduction study	+ Depression	+	Hypogonadism	-	+	

Diabetes mellitus (DM), diabetes insipidus (DI), optic atrophy (OA), sensorineural hearing loss (SNHL), neurological disorders (ND), psychiatric disorders (PD), urinary disorders (UD), endocrine disorders (ED), intestinal bleeding (IB), peptic ulcers (PU), platelet aggregation defects (PAD). Empty cells = data not available; * optic neuropathy rather than optic atrophy.

The clinical characteristics of patients with WS2 are often described together with those of patients with WS1, so it is not always easy to discriminate between the two forms of the disease. Some clinical manifestations of WS2 may be common to both type 1 and type 2, not allowing a precise clinical differential diagnosis between the two subtypes. Table 2 shows the clinical differences between WS1 and WS2.

### 3.12. Treatment

Unfortunately, there are currently no specific treatments that are able to restore ER function and prevent the complications that are caused by this disorder. There are also no treatments that are currently able to prevent the progression of WS.

Five therapeutic clinical trials for WS are currently available on http://clinicaltrial.gov, accessed on 15 December 2021. Among these studies, two evaluate the efficacy of sodium valproate (this study is currently in the recruitment phase), one the efficacy of dantrolene (this study is active and no longer recruiting), one the efficacy of sitagliptin alone and another one the efficacy of sitagliptin, deferiprone, acetylcysteine and metformin (these final two are studies from over two years ago from which we have no data for more than 2 years). These studies are mostly conduced on patients who are affected by WS1.

Calpain 2 calcium-mediated activation, a process that is negatively regulated by WS2, is involved in the regulation of cell death through caspase-3 activation. Studies show that calpain and mediators of its pathway could be therapeutic targets for WS and other ER stress diseases. Lu et al. tested several molecules that could modulate cytosolic calcium in a way that protects cells from ER stress and prevents cell death. Among the 73 chemical compounds that were tested, 8 could significantly reduce cell death (PARP inhibitor, dantrolene, NS398, pioglitazone, calpain III inhibitor, docosahexaenoic acid, rapamycin and GLP-1) [35].

Dantrolene is an inhibitor of ER-localized ryanodine receptors that is approved for the treatment of muscle spasticity and malignant hyperthermia. Dantrolene could reduce cellular stress in WS patients by preventing calcium’s passage into the cytosol and the occurrence of apoptosis that is induced by calpain activation. A phase Ib/IIa clinical trial has suggested that dantrolene improves pancreatic β-cell function by reducing cellular stress, but further studies are needed to confirm its efficacy in the treatment of Wolfram syndrome [36].

Some recent studies suggest that 4-phenylbutyrate (PBA) and valproate might reduce the ER stress and cell apoptosis in patients with WS1. PBA reduces ER stress in pancreatic β-cells by decreasing protein folding load and increasing insulin secretion. Valproate, widely used as an anticonvulsant and mood-stabilizing drug, has been reported to attenuate ER stress-induced apoptosis in neuronal, hepatocellular and pancreatic β-cells through an as yet unclear mechanism [37].

Glucagon-like peptide 1 (GLP-1) is an incretin hormone, which is secreted from the gut, that amplifies insulin secretion by stimulating cAMP and protein kinase A in response to a meal. Treatment with exenatide (GLP-1 receptor agonist) seems to improve beta-cell function and glycemic control, reducing the daily insulin requirement by 70% in WS2 patients. GLP-1 receptor agonists seem to improve mitochondrial function by preventing labile iron accumulation and increasing the response of insulin to stimulation, probably by up-regulating cAMP-stimulated insulin secretion [31]. Based on these findings, GLP-1 receptor agonists could be a therapeutic option for glycemic control in diabetic WS2 patients.

Additionally, the observation that some conditions that result in iron overload (such as Friedreich’s ataxia and WS2) develop diabetes underscores a link between iron metabolism and β-cell failure. Recent studies have shown that a treatment with a combination of deferiprone (a cell permeant iron chelator) and N-acetyl cysteine (a glutathione precursor) improved the structural repair of mitochondria and ER, reduced the mitochondrial labile iron and ROS levels and renewed insulin secretion. Furthermore, the authors demonstrated that the suppressed expression of the protein [2Fe-2S] NAF-1, encoded by the CISD2 gene, is associated with the development of ferroptosis-like characteristics in pancreatic cells. They showed that treatment with the ferroptosis inhibitor ferrostatin-1 improved the cellular growth of NAF-1-repressed pancreatic cells. The authors concluded that the reduction of the mitochondrial iron levels could be used as a therapeutic route for patients with WS2 within the in vitro study [38]. Human studies are needed in order to verify the efficacy of molecules that block ER stress-induced apoptosis in the treatment of WS2.

## 4. Discussion

WS is a rare and severe multisystem neurodegenerative disease, so many studies have described its clinical features; however, these studies are almost always case reports involving a few subjects. There are only two nationwide cross-sectional population-based studies describing the natural history, complications, prevalence and inheritance of WS1 syndrome. These studies, including data from 68 and 45 patients, showed, with slight differences in prevalence, the constant presence of the four cardinal symptoms (in both studies the prevalence of diabetes mellitus and optic atrophy was 100%) and variable association with other signs and symptoms of systemic involvement [7,39]. In contrast, there are no large studies that have collected information on the clinical spectrum of a large case series of WS2.

In the context of WS, it is even rarer to find studies of WS2. WS2 comes with overlapping clinical features, except for (almost always) the absence of diabetes insipidus and the presence of gastrointestinal ulcers and platelet aggregation deficiency.

It is not uncommon to observe gastrointestinal ulcers and bleeding tendency as the first manifestations of WS2, where diabetes is always present, with onset in school-age (8–9 years). In fact, in our case [10,15], the onset symptom of WS2, which preceded the onset of diabetes mellitus by years, was intestinal malrotation associated with intussusception requiring surgery [10]. However, we found no described cases of WS2 patients without nonautoimmune diabetes mellitus in the literature.

Deafness appears to be constant in WS, but its presentation is subtler in WS2 patients and requires audiometric testing in order to be detected. In some WS2 patients, deafness is present only at higher frequencies of sound [10].

In contrast, OA may be absent in some WS2 patients [10,15] who do, however, show optic neuropathy. DI appears to be anecdotal/partial in WS2 compared to WS1.

Bladder dysfunction is very common in WS1 and WS2 (~90%) with later progression to megacystis. This clinical feature is not usually present in the patients who are of pediatric age.

Neurological disorders are rare in WS2 pediatric patients and are described in the more advanced stages of life as a result of the progressive neurodegenerative disease. Psychiatric symptoms seem to be less frequent in WS2 than in WS1. However, it is worthwhile to consider that all chronic pathologies can cause mood depression, such as that which is described in some cases of WS2 in the adult age.

Wolfram syndrome can be confused with other genetic syndromes that share some clinical manifestations (i.e., neurological disorders), such as Friedreich’s ataxia, which is only sometimes associated with diabetes (8–32% of patients) [40]

In patients with Friedreich’s ataxia, hyperglycemia commonly develops approximately 15 years after the manifestation of neurological symptoms. This differs from patients with WS2, for whom diabetes is one of the first clinical manifestations.

WS negatively impacts the affected individuals’ quality of life and participation in various activities of daily living. These limitations tend to be more pronounced with advancing age and disease-related neurodegeneration (loss of vision or hearing, walking and balance problems, etc.), but may already be evident in patients of pediatric age. Currently, the first therapeutic approach in WS is to manage the clinical symptoms, particularly diabetes; but more attention should be given to occupational therapy, especially in pediatric patients, enhancing social interaction and identifying sports that are compatible with any visual, auditory or neurological difficulties [41].

The heterogeneity of the presentation of WS2 is likely due to the multifaceted role and almost ubiquitous distribution of the CISD2 protein, which is involved in numerous mechanisms of cell survival. In fact, CISD2 is involved in the regulatory mechanisms of calcium homeostasis, autophagy, ROS production and apoptosis. To date, there are no effective treatments available for WS2, but several clinical trials for drugs that are capable of blocking the cascade of cellular events that is mediated by CISD2 are in progress.

## 5. Conclusions

Pediatricians should generally suspect syndromic forms in which insulin-dependent non-autoimmune diabetes mellitus is present when the condition is associated with pathological manifestations that are affecting organs such as the eye and ear.

The rarest form of WS, WS2, is suspectable when, in addition to insulin-dependent non-autoimmune diabetes mellitus, other clinical manifestations such as bowel disease, visual changes due to optic atrophy/neuropathy or hematological defects are present. The platelet aggregation defect, as well as the presence of deafness at high frequencies, are not always clinically evident and should be investigated with specific diagnostic tests. Early diagnosis of WS is essential in order to prevent the development of long-term complications, improve the patients’ quality of life and morbidity-mortality and reduce the risk of transmission from parents to offspring through the use of proper genetic counselling.

Given the broad spectrum of clinical manifestations, a specific multidisciplinary approach is required. In cases of clinical suspicion of WS2, this approach should at least include ophthalmological, audiological, gastrointestinal and hematological evaluations. At the same time, during the follow-up, it may be necessary to include other specialist consultations such as neurological, urological and psychiatric ones.

In conclusion, non-autoimmune insulin-dependent DM, OA and SNHL represent the three cardinal elements of WS1. In contrast, peptic ulcers, bleeding tendency, non-autoimmune insulin-dependent DM, oculopathy and SNHL are the hallmarks of WS2. A more specific acronym PUBleeDMOD (denoting peptic ulcers, bleeding, diabetes mellitus, optic involvement and deafness) may be conceivable for WS2.

## Figures and Tables

**Figure 1 ijerph-19-00835-f001:**
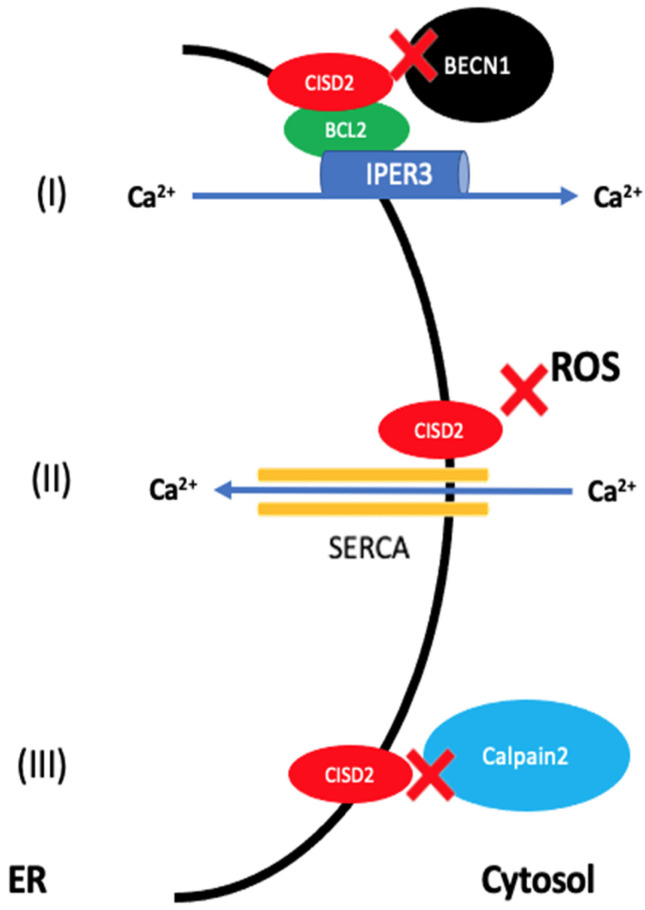
Main functions of CISD2 gene (modified from Shen ZQ et al.). (**I**) CISD2 forms a complex with BCL-2 that blocks BECN1-mediated autophagy. If CISD2 is deficient, BECN1-induced autophagy is uncontrolled. In addition, the CISD2-BCL-2 complex regulates mitochondrial Ca^2+^ homeostasis via IP3R. Mutations in CISD2 may impair the function of IP3R and thus calcium homeostasis. (**II**) CISD2 regulates reduction-oxidation mechanisms by preventing oxidation of SERCA proteins, which are essential for cytosolic Ca^2+^ transport in the ER. CISD2 deficiency reduces SERCA activity resulting in increased cytosolic Ca^2+^, increased ER stress and ROS production. (**III**) CISD2 inhibits apoptosis mediated by Calpain2 and caspase 3. In the absence of CISD2, activation of Calpain2 leads to increased apoptotic processes.

**Figure 2 ijerph-19-00835-f002:**
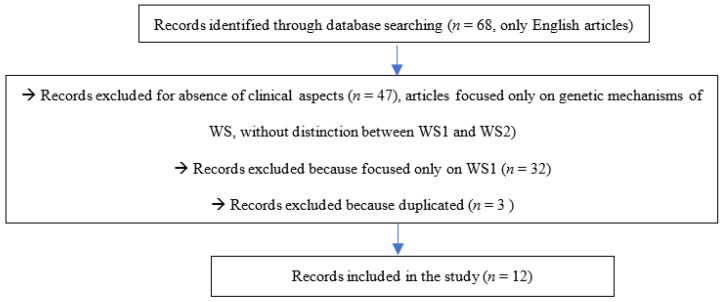
Article process selection.

**Table 2 ijerph-19-00835-t002:** Clinical differences between WS1 and WS2.

Clinical Features	WS1	WS2
Diabetes mellitus	Yes	Yes
Optic atrophy/neuropathy	Yes	Yes
Sensorineural deafness	Yes	Yes
Diabetes insipidus	Yes	No (sporadic cases described)
Other endocrinological abnormalities	Yes	No (sporadic cases described)
Neurological disorders	Yes	No (sporadic cases described)
Psychiatric symptoms	Yes	No (mood disorders described)
Genitourinary problems	Yes	Yes
Platelet aggregation defects	No	Yes
GI ulcers	No	Yes

## Data Availability

The datasets that were generated and/or analyzed during the current study are available from the corresponding author on reasonable request.

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
