# Peer review of "Wolfram Syndrome Type 2: A Systematic Review of a Not Easily Identifiable Clinical Spectrum"

_ijerph, 2022, doi:10.3390/ijerph19020835_

Round 1

Reviewer 1 Report

Current review about WS2 is well written and has a high impact on Wolfram syndrome field. Authors described the major clinical features of WS2 described in the literature. This review might raise awareness about WS2 and help pediatrician to diagnose WS2 more easily. 

Author Response

Reviewer 1

(x) English language and style are fine/minor spell check required

Current review about WS2 is well written and has a high impact on Wolfram syndrome field. Authors described the major clinical features of WS2 described in the literature. This review might raise awareness about WS2 and help pediatrician to diagnose WS2 more easily. 

Thanks to the reviewer for his comments. We did a further spell check and made the right changes

Reviewer 2 Report

This is a fine, nicely written, and quite a necessary review on WS2. Apart from small typing errors I have only a few comments- the discussion is repetitive and needs to be rewritten. Also, I want authors to elaborate on treatment options, as the main audience is probably doctors dealing with WS. Indeed, there is a report on GLP1 in Jewish WS2 patients and it is mentioned in this paper. As WS2 is a disorder of iron accumulation, are there any reports on iron chelators and their success or lack of it in the treatment of WS2? Authors should also elaborate on prospective drug options. There is more than one drug in clinical trials for WS1 (valproate, dantrolene, and GLP1 RA to my knowledge). Given the similarity of WS2 and WS1, would treatment options for WS1 be translatable to WS2?  Given the rarity of WS2, it might be beneficial to „team“ this disorder with other, similar disorders and repurpose the treatments from there. How similar is WS2 to other iron-related disorders? Symptoms of Friedreich’s ataxia (FA) are quite similar to WS, perhaps the severity and order of appearance of symptoms are slightly different. Is there a possibility and rationale that new treatments for FA might be effective also for WS2? I wich authors elaborate the treatment section a bit more.

The section about the function of CISD2 concentrates on ER stress, however, there are also reports demonstrating that CISD2 participates in the transport of iron-sulfur clusters to other proteins. In a way, this fact makes WS2 more similar to FA than to WS1. 

One very puzzling question regarding WS is its rarity. The frequency of heterozygous Wfs1 mutations is about 1% in the population. Thus, the frequency of WS should be in the order of 1/40 000. But it is much less (1/700 000). What is the frequency of heterozygous CISD2 mutations in the general population? Also, I wish the authors to provide a schematic drawing of CISD2 protein with known invalidating mutations indicated in it. Or, at the very least, make one more column in the table to include mutation(s) for respective patients/studies.

Is there a possibility for WS2 without diabetes mellitus? 

The blood abnormalities may be present also in WS1 patients. Are blood abnormalities found in WS2 patients similar to blood findings in other iron-related disorders?

Is anything known about exercise tolerance for WS2 patients? Many WS1 patients are heat intolerant. Are WS2 patients also heat intolerant?

Based on that, I recommend a minor revision for this otherwise fine review on WS2.

There are a few typos, maybe more:

Line 21: peptide ulcers must be peptic ulcers and throughout the text.

Line 37: atypical Friederich ataxia Must be atypical Friedreich's ataxia, also elsewhere

There are instances where „they“ must be „these“ or „there“. Please check the spelling carefully.

Author Response

Reviewer 2

English language and style are fine/minor spell check required

Thanks to the reviewer for all  his comments which allowed us to improve the article. We did a further spell check and made the right changes

  • This is a fine, nicely written, and quite a necessary review on WS2. Apart from small typing errors I have only a few comments- the discussion is repetitive and needs to be rewritten. Also, I want authors to elaborate on treatment options, as the main audience is probably doctors dealing with WS. Indeed, there is a report on GLP1 in Jewish WS2 patients and it is mentioned in this paper. As WS2 is a disorder of iron accumulation, are there any reports on iron chelators and their success or lack of it in the treatment of WS2? Authors should also elaborate on prospective drug options. There is more than one drug in clinical trials for WS1 (valproate, dantrolene, and GLP1 RA to my knowledge). Given the similarity of WS2 and WS1, would treatment options for WS1 be translatable to WS2? Given the rarity of WS2, it might be beneficial to „team“ this disorder with other, similar disorders and repurpose the treatments from there. How similar is WS2 to other iron-related disorders? Symptoms of Friedreich’s ataxia (FA) are quite similar to WS, perhaps the severity and order of appearance of symptoms are slightly different. Is there a possibility and rationale that new treatments for FA might be effective also for WS2? I wich authors elaborate the treatment section a bit more.
  • Thanks to the reviewer for the suggestions that let us to amliorate the article.
  • We enriched the section related to therapeutic options including all the suggestions of the reviewer and clinical trials currently in progress. There are no ongoing human trials for WS2. We also delved into the issue of iron accumulation and chelation therapy. We added the differences from Friedreich’s ataxia in Discussion section.

  • The section about the function of CISD2 concentrates on ER stress, however, there are also reports demonstrating that CISD2 participates in the transport of iron-sulfur clusters to other proteins. In a way, this fact makes WS2 more similar to FA than to WS1.

Tanks for the advice, we have highlighted the issue of iron accumulation and chelation therapy and the differences with FA.

  • One very puzzling question regarding WS is its rarity. The frequency of heterozygous WS1 mutations is about 1% in the population. Thus, the frequency of WS should be in the order of 1/40 000. But it is much less (1/700 000). What is the frequency of heterozygous CISD2 mutations in the general population? Also, I wish the authors to provide a schematic drawing of CISD2 protein with known invalidating mutations indicated in it. Or, at the very least, make one more column in the table to include mutation(s) for respective patients/studies.

Thank you for the question. It would be very interesting to know the prevalence of heterozygotes for WS2. However the prevalence of Wolfram syndrome (both 1 and 2) is 1/770000 in rare disease data collection platforms. This specific WS2 data is not known actually. Currently, the prevalence of Wolfram syndrome (both 1 and 2) is 1/770000 in rare disease data collection platforms. At present the specific prevalence of WS2 is probably even higher than we imagine.

WS2 should be investigated in not-autoimmune diabetic patients with bleeding ulcers and platelet abnormalities who may not be recognized as deaf since they suffer from sensorineural hearing loss on high frequencies and who may not be blind because they do not suffer from optic atrophy, patients where neurological manifestations are subtle and poorly recognized.

In addition, we included a figure of the CISD2 protein and its main mechanisms. We included a figure of the CISD2 protein and its main mechanisms and we also included a more column in the table to showe mutation(s) for respective patients/studies, where known.

  • Is there a possibility for WS2 without diabetes mellitus?
  • To the best of our knowledge no cases of WS2 have been described without diabetes, which remains one of the key elements for the diagnosis of this syndrome.

  • The blood abnormalities may be present also in WS1 patients. Are blood abnormalities found in WS2 patients similar to blood findings in other iron-related disorders?

The main hematologic abnormality described in WS2 patients,  key clinical element not recognized in WS1, is the bleeding tendency due to still unclear defects in the platelet aggregation. Furthermore, this mechanism seems to be different from that of other iron storage diseases.

  • Is anything known about exercise tolerance for WS2 patients? Many WS1 patients are heat Are WS2 patients also heat intolerant?
  • Thank you for your comment. we have not found any articles in the literature showing poor exercise or heat tolerance. however, due to neurodegeneration, normal social and sporting activities may be impaired in WS patients. we have highlighted this in the revised version of the paper.

Based on that, I recommend a minor revision for this otherwise fine review on WS2.

There are a few typos, maybe more:

Line 21: peptide ulcers must be peptic ulcers and throughout the text.

Line 37: atypical Friederich ataxia Must be atypical Friedreich's ataxia, also elsewhere

There are instances where „they“ must be „these“ or „there“. Please check the spelling carefully.

Thanks to the reviewer. We made the right changes as suggested.

Reviewer 3 Report

This paper presents a review of phenotypic characteristics of Wolfram Syndrome, especially stressing those for its recently established subtype WS2 led by a disease-causing mutation appearing in the CISD2 gene, with the aim of differentiating this new WS subtype from the classic form. An effort is given by collecting evidence from literature published from July 2001 to July 2021. The authors of this paper also describe the links between this new subtype and other related diseases. Overall, the authors basically provide a well-written form of text for a description of phenotypic characteristics of the Wolfram syndrome. In terms of its detailed description and the comparatively thorough collection of evidence of phenotypic characteristics in clinic, this paper deserves to be published in this journal. Before that, the authors should address the following concerns.

  1. In section Introduction, the tail of the last paragraph has two full stops.
  2. The authors should confirm whether the caption of Figure 1 is well integrated.
  3. A suggested revision in the 3rd paragraph in Section Result: "Still, the presence of some additional findings may guide to suspect the diagnosis 130 as defective platelet aggregation and peptic ulcers with a bleeding tendency." to "Still, the presence of some additional findings may guide the suspect of the diagnosis 130 as defective platelet aggregation and peptic ulcers with a bleeding tendency."
  1. The authors collected a total of 35 patients of WS2 and inferred no epidemiological data for WS2. Do the epidemiological data refer to a category of data including sequencing data? Among these collected findings, how many DNA- or RNA sequencing data sources are involved. In my view, although a detailed description of the clinical phenotypic characteristics of WS2 is necessarily needed, sources of sequencing data for related genes or other forms of data to be derived from the WS2 analysis are also quite important for researchers to study them further. I believe that this will collectively and greatly contribute to the data-driven analysis led by scientists in this field. I suggest that in addition to constructing tables 1 and 2, the authors should attempt to provide another table for inferring the data sources especially for sequencing data or related protein data with a description of how to access them (e.g., as in the second paragraph in section Discussion, what types of data those studies used to evaluate the 65 and 45 patients affected by WS?). I do reckon this will benefit computational scientists to further interpret the causing and other implications of this newly found subtype WS2. If this could not be possible to do considering lots of current situations, the authors should at least discuss to what degree this can be available, and infer and provide good reasons and proofs for this.

Author Response

Reviewer 3

This paper presents a review of phenotypic characteristics of Wolfram Syndrome, especially stressing those for its recently established subtype WS2 led by a disease-causing mutation appearing in the CISD2 gene, with the aim of differentiating this new WS subtype from the classic form. An effort is given by collecting evidence from literature published from July 2001 to July 2021. The authors of this paper also describe the links between this new subtype and other related diseases. Overall, the authors basically provide a well-written form of text for a description of phenotypic characteristics of the Wolfram syndrome. In terms of its detailed description and the comparatively thorough collection of evidence of phenotypic characteristics in clinic, this paper deserves to be published in this journal. Before that, the authors should address the following concerns.

Thanks to the reviewer for all  his comments which allowed us to improve the article. We did the right changes as follows.

  1. In section Introduction, the tail of the last paragraph has two full stops.

Thanks for the correction. Done

  1. The authors should confirm whether the caption of Figure 1 is well integrated.

Figure 1 is well integrated, thanks for the correction.

  1. A suggested revision in the 3rd paragraph in Section Result: "Still, the presence of some additional findings may guide to suspect the diagnosis as defective platelet aggregation and peptic ulcers with a bleeding tendency." to "Still, the presence of some additional findings may guide the suspect of the diagnosis  as defective platelet aggregation and peptic ulcers with a bleeding tendency."

We made the right corrections in the paragraph as suggested.

  1. The authors collected a total of 35 patients of WS2 and inferred no epidemiological data for WS2. Do the epidemiological data refer to a category of data including sequencing data? Among these collected findings, how many DNA- or RNA sequencing data sources are involved. In my view, although a detailed description of the clinical phenotypic characteristics of WS2 is necessarily needed, sources of sequencing data for related genes or other forms of data to be derived from the WS2 analysis are also quite important for researchers to study them further. I believe that this will collectively and greatly contribute to the data-driven analysis led by scientists in this field. I suggest that in addition to constructing tables 1 and 2, the authors should attempt to provide another table for inferring the data sources especially for sequencing data or related protein data with a description of how to access them (e.g., as in the second paragraph in section Discussion, what types of data those studies used to evaluate the 65 and 45 patients affected by WS?). I do reckon this will benefit computational scientists to further interpret the causing and other implications of this newly found subtype WS2. If this could not be possible to do considering lots of current situations, the authors should at least discuss to what degree this can be available, and infer and provide good reasons and proofs for this.

Thanks for this comment.

Regarding epidemiological data. We included a more column in the table to showe sequencing data for respective patients/studies, where known from literature. The prevalence of Wolfram syndrome (both 1 and 2) is 1/770000 in rare disease data collection platforms. The specific WS2 epidemiological data is not known actually.  At present the specific prevalence of WS2 is probably even higher than we imagine.

In Discussion section we added data regarding studies evaluating the 65 and 45 patients affected by WS.

We also believe that the reviewer has made some very good suggestions regarding being more precise on the sequencing data in the literature, however few are known compared to the published clinical cases of WGS2, and we have provided them.
